# Sliced Iterative Normalizing Flows

**Biwei Dai** [1]   **Uroš Seljak** [1] [2]

## Abstract

We develop an iterative (greedy) deep learning algorithm which is able to transform an arbitrary probability distribution function (PDF) into the target PDF. The model is based on iterative Optimal Transport of a series of 1D slices, matching on each slice the marginal PDF to the target. As special cases of this algorithm, we introduce two Sliced Iterative Normalizing Flows (SINF), which map from the data to the latent space (GIS) and vice versa (SIG). We show that SIG is able to generate high quality samples that match the GAN benchmarks. GIS obtains better results on small dataset density estimation tasks compared to the density trained NFs. SINF approach deviates significantly from the current DL paradigm, as it is greedy and does not use concepts such as minibatching, stochastic gradient descent and gradient back-propagation through deep layers.

## 1. Introduction

Latent variable generative models such as Normalizing Flows (NFs) (Rezende & Mohamed, 2015; Dinh et al., 2014; 2017; Kingma & Dhariwal, 2018), Variational AutoEncoders (VAEs) (Kingma & Welling, 2014; Rezende et al., 2014) and Generative Adversarial Networks (GANs) (Goodfellow et al., 2014; Radford et al., 2016) aim to model the distribution $p(x)$ of high-dimensional input data $x$ by introducing a mapping from a latent variable $z$ to $x$, where $z$ is assumed to follow a given prior distribution $\pi(z)$. These models usually parameterize the mapping using neural networks, and the training of these models typically consists of minimizing a dissimilarity measure between the model distribution and the target distribution.

In this work we adopt a different approach to build the map

---

[*]Equal contribution [1]Department of Physics, University of California, Berkeley, California, USA [2]Lawrence Berkeley National Laboratory, Berkeley, California, USA. Correspondence to: Biwei Dai <biwei@berkeley.edu>.

Third workshop on *Invertible Neural Networks, Normalizing Flows, and Explicit Likelihood Models* (ICML 2021). Copyright 2021 by the author(s).

from latent variable $z$ to data $x$. We approach this problem from the Optimal Transport (OT) point of view. OT studies whether the transport maps exist between two probability distributions, and if they do, how to construct the map to minimize the transport cost. We propose to decompose the high dimensional problem into a succession of 1D transport problems, where the OT solution is known. The mapping is iteratively augmented, and it has a NF structure that allows explicit density estimation and efficient sampling. We name the algorithm Sliced Iterative Normalizing Flow (SINF). Our objective function is inspired by the Wasserstein distance, which is defined as the minimal transport cost and has been widely used in the loss functions of generative models (Arjovsky & Bottou, 2017; Tolstikhin et al., 2018). We propose a new metric, max K-sliced Wasserstein distance, which enables the algorithm to scale well to high dimensions.

## 2. Background

The p-Wasserstein distance, $p \in [1, \infty)$, between two probability distributions $p_1$ and $p_2$ is defined as:

$$W_p(p_1, p_2) = \inf_{\gamma \in \Pi(p_1, p_2)} \left( \mathbb{E}_{(x,y) \sim \gamma} \left[ \|x - y\|^p \right] \right)^{\frac{1}{p}}, \quad (1)$$

where $\Pi(p_1, p_2)$ is the set of all possible joint distributions $\gamma(x, y)$ with marginalized distributions $p_1$ and $p_2$. In 1D the Wasserstein distance has a closed form solution via Cumulative Distribution Functions (CDFs), but this evaluation is intractable in high dimension. An alternative metric, the Sliced p-Wasserstein Distance (SWD) (Rabin et al., 2011; Bonneel et al., 2015), is defined as:

$$SW_p(p_1, p_2) = \left( \int_{\mathbb{S}^{d-1}} W_p^p(\mathcal{R}p_1(\cdot, \theta), \mathcal{R}p_2(\cdot, \theta)) d\theta \right)^{\frac{1}{p}}, \quad (2)$$

where $\mathbb{S}^{d-1}$ denotes the unit sphere $\theta_1^2 + \cdots \theta_n^2 = 1$ in $\mathbb{R}^d$, $d\theta$ is the normalized uniform measure on $\mathbb{S}^{d-1}$, and $\mathcal{R}$ denotes the Radon transform. The definition of Radon transform can be found in the appendix. For a given $\theta$, the function $(\mathcal{R}p)(\cdot, \theta) : \mathbb{R} \to \mathbb{R}$ is essentially the slice (or projection) of $p(x)$ on axis $\theta$.

The SWD can be calculated by approximating the high dimensional integral with Monte Carlo samples. However, in high dimensions a large number of projections is required

to accurately estimate SWD. This motivates to use the maximum Sliced p-Wasserstein Distance (max SWD):

$$\text{max-}SW_p(p_1, p_2) = \max_{\theta \in \mathbb{S}^{d-1}} W_p(\mathcal{R}p_1(\cdot, \theta), \mathcal{R}p_2(\cdot, \theta)), \tag{3}$$

which is the maximum of the Wasserstein distance of the 1D marginalized distributions of all possible directions.

# 3. Sliced Iterative Normalizing Flows

We consider the general problem of building a NF that maps an arbitrary PDF $p_1(x)$ to another arbitrary PDF $p_2(x)$ of the same dimensionality. We firstly introduce our objective function in Section 3.1. The general SINF algorithm is presented in Section 3.2. We then consider the special cases of $p_1$ and $p_2$ being standard Normal distributions in Section 3.3 and Section 3.4, respectively.

## 3.1. Maximum K-sliced Wasserstein distance

We generalize the idea of maximum SWD and propose maximum K-Sliced p-Wasserstein Distance (max K-SWD):

$$\text{max-}K\text{-}SW_p(p_1, p_2) = \max_{\{\theta_1, \cdots, \theta_K\} \text{ orthonormal}}$$

$$\left( \frac{1}{K} \sum_{k=1}^{K} W_p^p((\mathcal{R}p_1)(\cdot, \theta_k), (\mathcal{R}p_2)(\cdot, \theta_k)) \right)^{\frac{1}{p}}. \tag{4}$$

In this work we fix $p = 2$. The max K-SWD defines K orthogonal axes $\{\theta_1, \cdots, \theta_K\}$ where the marginal distributions of $p_1$ and $p_2$ are most different, providing a natural choice for performing 1D marginal matching in our algorithm (see Section 3.2). The proof that max K-SWD is a proper distance and the details of its estimation are provided in the appendix.

## 3.2. Proposed SINF algorithm

The SINF algorithm is based on iteratively matching the 1D marginalized distribution of $p_1$ to $p_2$. This is motivated by the inverse Radon Transform (see Appendix) and Cramér-Wold theorem, which suggest that matching the high dimensional distributions is equivalent to matching the 1D slices on all possible directions, decomposing the high dimensional problem into a series of 1D problems. Given a set of i.i.d. samples $X$ drawn from $p_1$, in each iteration, a set of 1D marginal transformations $\{\Psi_k\}_{k=1}^{K}$ [1] ($K \leq d$ where $d$ is the dimensionality of the dataset) are applied to the samples on orthogonal axes $\{\theta_k\}_{k=1}^{K}$ to match the 1D marginalized PDF of $p_2$ along those axes. Let $A = [\theta_1, \cdots, \theta_K]$ be the

---

[1]**Notation definition**: In this paper we use $l$, $k$, $j$ and $m$ to represent different iterations of the algorithm, different axes $\theta_k$, different gradient descent iterations of max K-SWD calculation (see Algorithm 2), and different knots in the spline functions of 1D transformation, respectively.

---

**Algorithm 1** Sliced Iterative Normalizing Flow

**Input:** $\{x_i \sim p_1\}_{i=1}^{N}$, $\{y_i \sim p_2\}_{i=1}^{N}$, $K$, number of iteration $L_{\text{iter}}$
**for** $l = 1$ **to** $L_{\text{iter}}$ **do**
    $\theta_1, \cdots, \theta_K = \arg\max$ K-SWD$(x_i, y_i, K)$
    **for** $k = 1$ **to** $K$ **do**
        Compute $\hat{x}_i = \theta_k \cdot x_i$ and $\hat{y}_i = \theta_k \cdot y_i$ for each $i$
        $\tilde{x}_m = \text{quantiles}(\text{PDF}(\hat{x}_i))$
        $\tilde{y}_m = \text{quantiles}(\text{PDF}(\hat{y}_i))$
        $\Psi_{l,k} = \text{RationalQuadraticSpline}(\tilde{x}_m, \tilde{y}_m)$
    **end for**
    $\Psi_l = [\Psi_1, \cdots, \Psi_K]$, $A_l = [\theta_1, \cdots, \theta_K]$
    Update $x_i = x_i - A_l A_l^T x_i + A_l \Psi_l(A_l^T x_i)$
**end for**

---

*Table 1.* Comparison between SIG and GIS

| Model | SIG | GIS |
|---|---|---|
| Initial PDF $p_1$ | Gaussian | $p_{\text{data}}$ |
| Final PDF $p_2$ | $p_{\text{data}}$ | Gaussian |
| Training | Iteratively maps Gaussian to $p_{\text{data}}$ | Iteratively maps $p_{\text{data}}$ to Gaussian |
| NF structure | Yes | Yes |
| Advantage | Good samples | Good density estimation |

---

weight matrix ($A^T A = I_K$), the transformation at iteration $l$ of samples $X_l$ can be written as

$$X_{l+1} = A_l \Psi_l(A_l^T X_l) + X_l^{\perp}, \tag{5}$$

where $X_l^{\perp} = X_l - A_l A_l^T X_l$. $\Psi_l = [\Psi_{l1}, \cdots, \Psi_{lK}]^T$ is the marginal mapping of each dimension of $A_l^T X_l$, and its components are required to be monotonic and differentiable. The Inverse and Jacobian determinant of transformation 5 can be easily evaluated (see appendix).

The weight matrix $A_l$ and the marginal transformations $\Psi_l$ are determined by iteratively minimizing the max K-SWD (Equation 4) between the transformed $p_1$ and $p_2$. Specifically, we propose to iteratively solving for the orthogonal axes $\{\theta_1, \cdots, \theta_K\}$ in max K-SWD, and then apply 1D marginal matching on those axes to minimize max K-SWD.

Let $p_{1,l}$ be the transformed $p_1$ at iteration $l$. The $k$th component of $\Psi_l$, $\Psi_{l,k}$, maps the 1D marginalized PDF of $p_{1,l}$ to $p_2$ and has an OT solution:

$$\Psi_{l,k}(x) = F_k^{-1}(G_{l,k}(x)), \tag{6}$$

where $G_{l,k}(x) = \int_{-\infty}^{x} (\mathcal{R}p_{1,l})(t, \theta_k)dt$ and $F_k(x) = \int_{-\infty}^{x} (\mathcal{R}p_2)(t, \theta_k)dt$ are the CDFs of $p_{1,l}$ and $p_2$ on axis $\theta_k$, respectively. The CDFs can be estimated using the quantiles of the samples (in SIG Section 3.3), or using Kernel Density

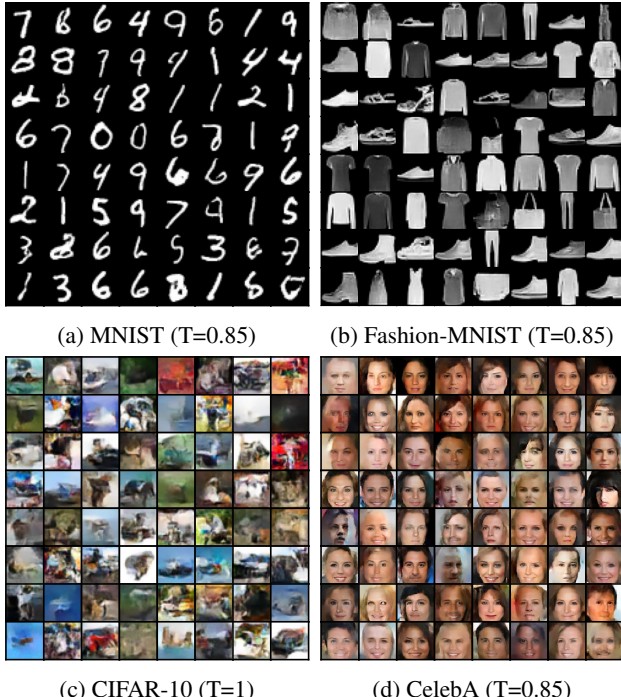

(a) MNIST (T=0.85)     (b) Fashion-MNIST (T=0.85)

(c) CIFAR-10 (T=1)     (d) CelebA (T=0.85)

*Figure 1.* Random samples from SIG.

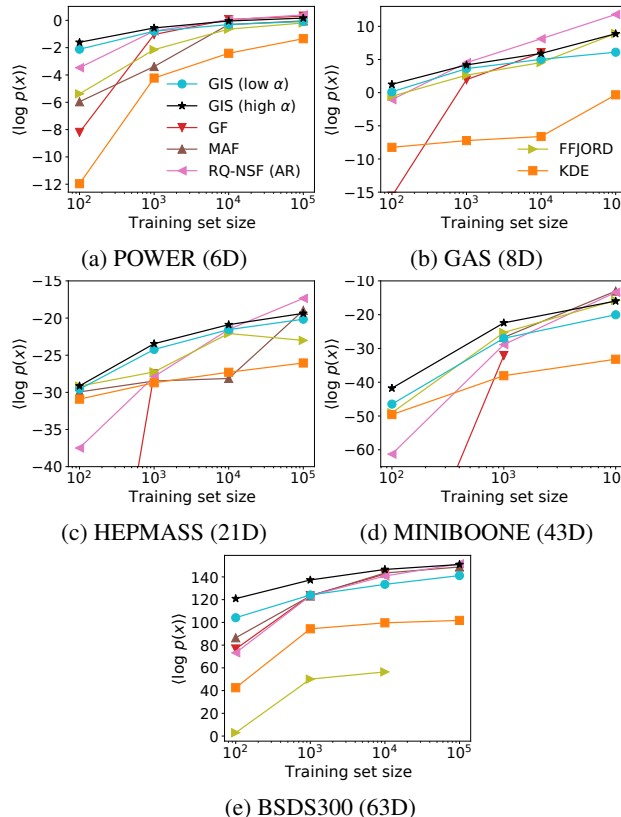

(a) POWER (6D)     (b) GAS (8D)

(c) HEPMASS (21D)     (d) MINIBOONE (43D)

(e) BSDS300 (63D)

*Figure 2.* Density estimation versus training set size. The legend in panel (a) applies to other panels as well. Higher is better: at 100-1000 training size GIS has the best performance in all cases.

Estimation (KDE, in GIS Section 3.4). Equation 6 is monotonic and therefore invertible. We choose to parametrize it with monotonic rational quadratic splines (Gregory & Delbourgo, 1982; Durkan et al., 2019). Details about the splines are shown in the appendix.

The proposed algorithm iteratively minimizes the max K-SWD between the transformed $p_1$ and $p_2$. The orthonomal vectors $\{\theta_1, \cdots, \theta_K\}$ specify $K$ axes along which the marginalized PDF between $p_{1,l}$ and $p_2$ are most different, thus maximizes the gain at each iteration and improves the efficiency of the algorithm. The model is able to converge with two orders of magnitude fewer iterations than random axes, and it also leads to better sample quality. This is because as the dimensionality $d$ grows, the number of slices $(\mathcal{R}p)(\cdot, \theta)$ required to approximate $p(x)$ using inverse Radon formula scales as $L^{d-1}$ (Kolouri et al., 2015), where $L$ is the number of slices needed to approximate a similar smooth 2D distribution. Therefore, if $\theta$ are randomly chosen, it takes a large number of iterations to converge in high dimensions due to the curse of dimensionality. Our objective function reduces the curse of dimensionality in high dimensions by identifying the most relevant directions.

Unlike KL-divergence which is invariant under flow transformations, max K-SWD is different in data space and in latent space. Therefore the direction of building the flow model is of key importance. In the next two sections we discuss two different ways of building the flow, which are good at sample generation and density estimation, respectively.

## 3.3. Sliced Iterative Generator (SIG)

For Sliced Iterative Generator (SIG) $p_1$ is a standard Normal distribution, and $p_2$ is the target distribution. The model iteratively maps the Normal distribution to the target distribution using 1D slice transformations. Specifically, one firstly draw a set of samples from the standard Normal distribution, and then iteratively updates the samples following Equation 5. SIG directly minimizes the max K-SWD between the generated distribution and the target distribution, and is able to generate high quality samples.

## 3.4. Gaussianizing Iterative Slicing (GIS)

For Gaussianizing Iterative Slicing (GIS) $p_1$ is the target distribution and $p_2$ is a standard Normal distribution. The model iteratively Gaussianizes the target distribution, and the mapping is learned in the reverse direction of SIG. In GIS the max K-SWD between latent data and the Normal distribution is minimized, thus the model performs well in density estimation, even though its objective is not $p(x)$. A comparison between SIG and GIS is shown in Table 1.

*Table 2.* FID scores on different datasets (lower is better). The errors are generally smaller than the differences.

| | Method | MNIST | Fashion | CIFAR-10 | CelebA |
|---|---|---|---|---|---|
| iterative | SWF | 225.1 | 207.6 | - | - |
| | SIG ($T = 1$) (this work) | **4.5** | **13.7** | 66.5 | 37.3 |
| adversarial training | Flow-GAN (ADV) | 155.6 | 216.9 | 71.1 | - |
| | WGAN | 6.7 | 21.5 | **55.2** | 41.3 |
| | WGAN GP | 20.3 | 24.5 | 55.8 | **30.0** |
| | Best default GAN | $\sim 10$ | $\sim 32$ | $\sim 70$ | $\sim 48$ |
| AE based | SWAE(Wu et al., 2019) | - | - | 107.9 | 48.9 |
| | SWAE(Kolouri et al., 2018) | 29.8 | 74.3 | 141.9 | 53.9 |
| | CWAE | 23.6 | 57.1 | 120.0 | 49.7 |
| | PAE | - | 28.0 | - | 49.2 |
| | two-stage VAE | 12.6 | 29.3 | 96.1 | 44.4 |

*Table 3.* Averaged training time of different NF models on small datasets ($N_{\mathrm{train}} = 100$) measured in seconds. All the models are tested on both a cpu and a K80 gpu, and the faster results are reported here (the results with * are run on gpus.). P: POWER, G: GAS, H: HEPMASS, M: MINIBOONE, B: BSDS300.

| Method | P | G | H | M | B |
|---|---|---|---|---|---|
| GIS (low $\alpha$) | 0.53 | 1.0 | 0.63 | 3.5 | 7.4 |
| GIS (high $\alpha$) | 6.8 | 9.4 | 7.3 | 44.1 | 69.1 |
| GF | 113* | 539* | 360* | 375* | 122* |
| MAF | 18.4 | -[1] | 10.2 | -[1] | 32.1 |
| FFJORD | 1051 | 1622 | 1596 | 499* | 4548* |
| RQ-NSF (AR) | 118 | 127 | 55.5 | 38.9 | 391 |

[1] Training failures.

# 4. Experiments

## 4.1. Generative modeling of images

We evaluate SIG as a generative model of images using the following 4 datasets: MNIST (LeCun et al., 1998), Fashion-MNIST (Xiao et al., 2017), CIFAR-10 (Krizhevsky et al., 2009) and Celeb-A (Liu et al., 2015). In Figure 1 we show samples of these four datasets. For MNIST, Fashion-MNIST and CelebA dataset we show samples from the model with reduced temperature $T = 0.85$ (i.e., sampling from a Gaussian distribution with standard deviation $T = 0.85$ in latent space), which slightly improves the sample quality (Parmar et al., 2018; Kingma & Dhariwal, 2018). We report the final FID score (calculated using temperature T=1) in Table 2, where we compare our results with similar algorithms SWF (Liutkus et al., 2019) and Flow-Gan (ADV) (Grover et al., 2018). We also list the FID scores of some other generative models for comparison, including models using slice-based distance SWAEs (Wu et al., 2019; Kolouri et al., 2018) and CWAE (Knop et al., 2018), Wasserstein GAN models

(Arjovsky et al., 2017; Gulrajani et al., 2017), and other GANs and AE-based models PAE (Böhm & Seljak, 2020) and two-stage VAE (Dai & Wipf, 2019; Xiao et al., 2019). We notice that previous iterative algorithms were unable to produce good samples on high dimensional image datasets. In contrast, SIG obtains the best FID scores on MNIST and Fashion-MNIST, while on CIFAR-10 and CelebA it also outperforms similar algorithms and AE-based models, and gets comparable results with GANs.

## 4.2. Density estimation $p(x)$ of small datasets

We perform density estimation with GIS on four UCI datasets (Lichman et al., 2013) and BSDS300 (Martin et al., 2001). The data preprocessing of UCI datasets and BSDS300 follows Papamakarios et al. (2017). We vary the size of the training set $N_{\mathrm{train}}$ from $10^2$ to $10^5$ to test the model performance on a wide range of dataset size. For GIS we consider two hyperparameter settings: large regularization $\alpha$ (see appendix for more details) for better $\log p$ performance, and small regularization $\alpha$ for faster training. In Figure 2 we compare GIS to other NF models GF (Meng et al., 2020), FFJORD (Grathwohl et al., 2019), MAF (Papamakarios et al., 2017) and RQ-NSF (AR)(Durkan et al., 2019), as well as KDE. Some non-GIS NF models collapsed during training or used more memory than our GPU, and are not shown in the plot. The results in Figure 2 show that GIS is more stable compared to other NFs and outperforms them on small training sets. This highlights that GIS is less sensitive to hyper-parameter optimization and achieves good performance out of the box. GIS training time varies with data size, but is generally lower than other NFs for small training sets. We report the training time for 100 training data in Table 3. GIS with small regularization $\alpha$ requires significantly less time than other NFs, while still outperforming them at 100 training size.

# 5. Conclusions

We introduce sliced iterative normalizing flow (SINF) that iteratively transforms data distribution to a Gaussian (GIS) or the other way around (SIG) using OT. To the best of our knowledge, SIG is the first greedy deep learning algorithm that is competitive with the SOTA generators in high dimensions, while GIS achieves comparable results on density estimation with current NF models, but is more stable, faster to train, and achieves higher $p(x)$ when trained on small training sets even though it does not train on $p(x)$. SINF has very few hyperparameters, and is very insensitive to their choice. SINF is related to several previous models, which is discussed in Appendix. SINF has deep neural network architecture, but its approach deviates significantly from the current Deep Learning paradigm, as it does not use concepts such as mini-batching, stochastic gradient descent and gradient back-propagation through deep layers. SINF is an existence proof that greedy Deep Learning without these ingredients can be SOTA for modern high dimensional ML applications. SINF may be of particular interest in applications where robustness, insensitivity to hyperparameters, small data size, and speed are of primary importance.

# Acknowledgements

We thank He Jia for providing his code on Iterative Gaussianization, and for helpful discussions. We thank Vanessa Boehm and Jascha Sohl-Dickstein for comments on the manuscript. This material is based upon work supported by the National Science Foundation under Grant Numbers 1814370 and NSF 1839217, and by NASA under Grant Number 80NSSC18K1274.

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

# Sliced Iterative Normalizing Flows

**SUPPLEMENTARY DOCUMENT**

## A. Radon transform

Let $\mathbb{L}^1(X)$ be the space of absolute integrable functions on $X$. The Radon transform $\mathcal{R} : \mathbb{L}^1(\mathbb{R}^d) \to \mathbb{L}^1(\mathbb{R} \times \mathbb{S}^{d-1})$ is defined as

$$(\mathcal{R}p)(t, \theta) = \int_{\mathbb{R}^d} p(x)\delta(t - \langle x, \theta\rangle)dx, \qquad (7)$$

where $\mathbb{S}^{d-1}$ denotes the unit sphere $\theta_1^2 + \cdots \theta_n^2 = 1$ in $\mathbb{R}^d$, $\delta(\cdot)$ is the Dirac delta function, and $\langle \cdot, \cdot \rangle$ is the standard inner product in $\mathbb{R}^d$. For a given $\theta$, the function $(\mathcal{R}p)(\cdot, \theta) : \mathbb{R} \to \mathbb{R}$ is essentially the slice (or projection) of $p(x)$ on axis $\theta$.

Note that the Radon transform $\mathcal{R}$ is invertible. Its inverse, also known as the filtered back-projection formula, is given by (Helgason, 2010; Kolouri et al., 2019)

$$\mathcal{R}^{-1}((\mathcal{R}p)(t, \theta))(x) = \int_{\mathbb{S}^{n-1}} ((\mathcal{R}p)(\cdot, \theta) * h)(\langle x, \theta\rangle)d\theta, \qquad (8)$$

where $*$ is the convolution operator, and the convolution kernel $h$ has the Fourier transform $\hat{h}(k) = c|k|^{d-1}$. The inverse Radon transform provides a practical way to reconstruct the original function $p(x)$ using its 1D slices $(\mathcal{R}p)(\cdot, \theta)$, and is widely used in medical imaging. This inverse formula implies that if the 1D slices of two functions are the same in all axes, these two functions are identical. This is also known as Carmér-Wold theorem (Cramér & Wold, 1936).

## B. max K-SWD

The optimization in max K-SWD is performed under the constraints that $\{\theta_1, \cdots, \theta_K\}$ are orthonormal vectors, or equivalently, $A^T A = I_K$ where $A = [\theta_1, \cdots, \theta_K]$ is the matrix whose i-th column vector is $\theta_i$. Mathematically, the set of all possible $A$ matrices is called Stiefel Manifold $V_K(\mathbb{R}^d) = \{A \in \mathbb{R}^{d \times K} : A^T A = I_K\}$. As suggested by Tagare (2011), the optimization of matrix $A$ can be performed by doing gradient ascent on the Stiefel Manifold:

$$A_{(j+1)} = \left(I_d + \frac{\tau}{2}B_{(j)}\right)^{-1}\left(I_d - \frac{\tau}{2}B_{(j)}\right)A_{(j)}, \quad (9)$$

where $A_{(j)}$ is the weight matrix at gradient descent iteration $j$ (which is different from the iteration $l$ of the algorithm), $\tau$ is the learning rate, which is determined by backtracking line search, $B = GA^T - AG^T$, and $G$ is the negative

gradient matrix $G = [-\frac{\partial \mathcal{F}}{\partial A_{p,q}}] \in \mathbb{R}^{d \times K}$. Equation 9 has the properties that $A_{(j+1)} \in V_K(\mathbb{R}^d)$, and that the tangent vector $\frac{dA_{(j+1)}}{d\tau}|_{\tau=0}$ is the projection of gradient $[\frac{\partial \mathcal{F}}{\partial A_{p,q}}]$ onto $T_{A_{(j)}}(V_K(\mathbb{R}^d))$ (the tangent space of $V_K(\mathbb{R}^d)$ at $A_{(j)}$) under the canonical inner product (Tagare, 2011).

However, Equation 9 requires the inversion of a $d \times d$ matrix, which is computationally expensive in high dimensions. The matrix inversion can be simplified using the Sherman-Morrison-Woodbury formula, which results in the following equation (Tagare, 2011):

$$A_{(j+1)} = A_{(j)} - \tau U_{(j)}(I_{2K} + \frac{\tau}{2}V_{(j)}^T U_{(j)})^{-1}V_{(j)}^T A_{(j)}, \qquad (10)$$

where $U = [G, A]$ (the concatenation of columns of $G$ and $W$) and $V = [A, -G]$. Equation 10 only involves the inversion of a $2K \times 2K$ matrix, where $K$ is the number of axes to apply marginal transformation in each iteration. For high dimensional data (e.g. images), we use a relatively small $K$ to avoid the inversion of large matrices. A large $K$ leads to faster training, but one would converge to similar results with a small $K$ using more iterations.

The procedure for estimating max K-SWD and $A$ is summarized in Algorithm 2.

**Proposition 1.** *Let $P_p(\Omega)$ be the set of Borel probability measures with finite p'th moment on metric space $(\Omega, d)$. The maximum K-sliced p-Wasserstein distance is a metric over $P_p(\Omega)$.*

*Proof.* We firstly prove the triangle inequality. Let $\mu_1$, $\mu_2$ and $\mu_3$ be probability measures in $P_p(\Omega)$ with probability density function $p_1$, $p_2$ and $p_3$, respectively. Let $\{\theta_1^*, \cdots, \theta_K^*\} = \arg\max_{\{\theta_1, \cdots, \theta_K\}\text{ orthonormal}}$

**Algorithm 2** max K-SWD

---

**Input:** $\{x_i \sim p_1\}_{i=1}^N$, $\{y_i \sim p_2\}_{i=1}^N$, $K$, order $p$, max iteration $J_{\text{maxiter}}$

Randomly initialize $A \in V_K(\mathbb{R}^d)$

**for** $j = 1$ **to** $J_{\text{maxiter}}$ **do**

  Initialize $D = 0$

  **for** $k = 1$ **to** $K$ **do**

    $\theta_k = A[:, k]$

    Compute $\hat{x}_i = \theta_k \cdot x_i$ and $\hat{y}_i = \theta_k \cdot y_i$ for each $i$

    Sort $\hat{x}_i$ and $\hat{x}_j$ in ascending order s.t. $\hat{x}_{i[n]} \leq \hat{x}_{i[n+1]}$ and $\hat{y}_{j[n]} \leq \hat{y}_{j[n+1]}$

    $D = D + \frac{1}{KN} \sum_{i=1}^N |\hat{x}_{i[n]} - \hat{y}_{j[n]}|^p$

  **end for**

  $G = [-\frac{\partial D}{\partial A_{i,j}}]$, $U = [G, A]$, $V = [A, -G]$

  Determine learning rate $\tau$ with backtracking line search

  $A = A - \tau U(I_{2K} + \frac{\tau}{2} V^T U)^{-1} V^T A$

  **if** $A$ has converged **then**

    Early stop

  **end if**

**end for**

**Output:** $D^{\frac{1}{p}} \approx \max\text{-}K\text{-}SW_p$, $A \approx [\theta_1, \cdots, \theta_K]$

---

$\left( \frac{1}{K} \sum_{k=1}^K W_p^p((\mathcal{R}p_1)(\cdot, \theta_k), (\mathcal{R}p_3)(\cdot, \theta_k)) \right)^{\frac{1}{p}}$; then

$$\max\text{-}K\text{-}SW_p(p_1, p_3)$$
$$= \max_{\{\theta_1, \cdots, \theta_K\} \text{ orthonormal}} \left( \frac{1}{K} \sum_{k=1}^K W_p^p((\mathcal{R}p_1)(\cdot, \theta_k), (\mathcal{R}p_3)(\cdot, \theta_k)) \right)^{\frac{1}{p}}$$
$$= \left( \frac{1}{K} \sum_{k=1}^K W_p^p((\mathcal{R}p_1)(\cdot, \theta_k^*), (\mathcal{R}p_3)(\cdot, \theta_k^*)) \right)^{\frac{1}{p}}$$
$$\leq \left( \frac{1}{K} \sum_{k=1}^K [W_p((\mathcal{R}p_1)(\cdot, \theta_k^*), (\mathcal{R}p_2)(\cdot, \theta_k^*)) + W_p((\mathcal{R}p_2)(\cdot, \theta_k^*), (\mathcal{R}p_3)(\cdot, \theta_k^*))]^p \right)^{\frac{1}{p}}$$
$$\leq \left( \frac{1}{K} \sum_{k=1}^K W_p^p((\mathcal{R}p_1)(\cdot, \theta_k^*), (\mathcal{R}p_2)(\cdot, \theta_k^*)) \right)^{\frac{1}{p}} \tag{11}$$
$$+ \left( \frac{1}{K} \sum_{k=1}^K W_p^p((\mathcal{R}p_2)(\cdot, \theta_k^*), (\mathcal{R}p_3)(\cdot, \theta_k^*)) \right)^{\frac{1}{p}}$$
$$\leq \max_{\{\theta_1, \cdots, \theta_K\} \text{ orthonormal}} \left( \frac{1}{K} \sum_{k=1}^K W_p^p((\mathcal{R}p_1)(\cdot, \theta_k), (\mathcal{R}p_2)(\cdot, \theta_k)) \right)^{\frac{1}{p}}$$
$$+ \max_{\{\theta_1, \cdots, \theta_K\} \text{ orthonormal}} \left( \frac{1}{K} \sum_{k=1}^K W_p^p((\mathcal{R}p_2)(\cdot, \theta_k), (\mathcal{R}p_3)(\cdot, \theta_k)) \right)^{\frac{1}{p}}$$
$$= \max\text{-}K\text{-}SW_p(p_1, p_2) + \max\text{-}K\text{-}SW_p(p_2, p_3),$$

where the first inequality comes from the triangle inequality of Wasserstein distance, and the second inequality follows Minkowski inequality. Therefore $\max\text{-}K\text{-}SW_p$ satisfies the triangle inequality.

Now we prove the identity of indiscernibles. For any probability measures $\mu_1$ and $\mu_2$ in $P_p(\Omega)$ with probability density function $p_1$ and $p_2$, let $\hat{\theta} = \arg\max_{\theta \in \mathbb{S}^{d-1}} W_p((\mathcal{R}p_1)(\cdot, \theta), (\mathcal{R}p_2)(\cdot, \theta))$, and $\{\theta_1^*, \cdots, \theta_K^*\} = \arg\max_{\{\theta_1, \cdots, \theta_K\} \text{ orthonormal}} \left( \frac{1}{K} \sum_{k=1}^K W_p^p((\mathcal{R}p_1)(\cdot, \theta_k), (\mathcal{R}p_2)(\cdot, \theta_k)) \right)^{\frac{1}{p}}$, we have

$$\max\text{-}K\text{-}SW_p(p_1, p_2)$$
$$= \left( \frac{1}{K} \sum_{k=1}^K W_p^p((\mathcal{R}p_1)(\cdot, \theta_k^*), (\mathcal{R}p_2)(\cdot, \theta_k^*)) \right)^{\frac{1}{p}}$$
$$\leq \left( \frac{1}{K} \sum_{k=1}^K W_p^p((\mathcal{R}p_1)(\cdot, \hat{\theta}), (\mathcal{R}p_2)(\cdot, \hat{\theta})) \right)^{\frac{1}{p}} \tag{12}$$
$$= W_p((\mathcal{R}p_1)(\cdot, \hat{\theta}), (\mathcal{R}p_2)(\cdot, \hat{\theta}))$$
$$= \max\text{-}SW_p(p_1, p_2).$$

On the other hand, let $\{\hat{\theta}, \tilde{\theta}_2, \cdots, \tilde{\theta}_K\}$ be a set of orthonormal vectors in $\mathbb{S}^{d-1}$ where the first element is $\hat{\theta}$, we have

$$\max\text{-}K\text{-}SW_p(p_1, p_2)$$
$$= \left( \frac{1}{K} \sum_{k=1}^K W_p^p((\mathcal{R}p_1)(\cdot, \theta_k^*), (\mathcal{R}p_2)(\cdot, \theta_k^*)) \right)^{\frac{1}{p}}$$
$$\geq \left( \frac{1}{K} W_p^p((\mathcal{R}p_1)(\cdot, \hat{\theta}), (\mathcal{R}p_2)(\cdot, \hat{\theta})) + \frac{1}{K} \sum_{k=2}^K W_p^p((\mathcal{R}p_1)(\cdot, \tilde{\theta}_k), (\mathcal{R}p_2)(\cdot, \tilde{\theta}_k)) \right)^{\frac{1}{p}} \tag{13}$$
$$\geq \left( \frac{1}{K} W_p^p((\mathcal{R}p_1)(\cdot, \hat{\theta}), (\mathcal{R}p_2)(\cdot, \hat{\theta})) \right)^{\frac{1}{p}}$$
$$= (\frac{1}{K})^{\frac{1}{p}} \max\text{-}SW_p(p_1, p_2).$$

Therefore we have $(\frac{1}{K})^{\frac{1}{p}} \max\text{-}SW_p(p_1, p_2) \leq \max\text{-}K\text{-}SW_p(p_1, p_2) \leq \max\text{-}SW_p(p_1, p_2)$. Thus $\max\text{-}K\text{-}SW_p(p_1, p_2) = 0 \Leftrightarrow \max\text{-}SW_p(p_1, p_2) = 0 \Leftrightarrow \mu_1 = \mu_2$, where we use the non-negativity and identity of indiscernibles of $\max\text{-}SW_p$.

Finally, the symmetry of $\max\text{-}K\text{-}SW_p$ can be proven using

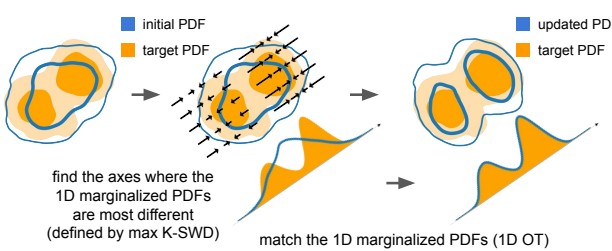

find the axes where the
1D marginalized PDFs
are most different
(defined by max K-SWD)

match the 1D marginalized PDFs (1D OT)

*Figure 3.* Illustration of 1 iteration of SINF algorithm with $K = 1$.

the fact that p-Wasserstein distance is symmetric:

$$\max\text{-}K\text{-}SW_p(p_1, p_2)$$

$$= \left( \frac{1}{K} \sum_{k=1}^{K} W_p^p((\mathcal{R}p_1)(\cdot, \theta_k^*), (\mathcal{R}p_2)(\cdot, \theta_k^*)) \right)^{\frac{1}{p}}$$

$$= \left( \frac{1}{K} \sum_{k=1}^{K} W_p^p((\mathcal{R}p_2)(\cdot, \theta_k^*), (\mathcal{R}p_1)(\cdot, \theta_k^*)) \right)^{\frac{1}{p}}$$

$$= \max\text{-}K\text{-}SW_p(p_2, p_1). \qquad (14)$$

$\square$

## C. More details of SINF

We show an illustration of the SINF algorithm in Figure 3.

### C.1. Inverse and Jacobian determinant

The SINF transformation of Equation 5 can be easily inverted:

$$X_l = A_l \Psi_l^{-1}(A_l^T X_{l+1}) + X_l^\perp, \qquad (15)$$

where $X_l^\perp = X_l - A_l A_l^T X_l = X_{l+1} - A_l A_l^T X_{l+1}$. The Jacobian determinant of the transformation is also efficient to calculate:

$$\det(\frac{\partial X_{l+1}}{\partial X_l}) = \prod_{k=1}^{K} \frac{d\Psi_{lk}(x)}{dx}. \qquad (16)$$

*Proof of Equation 16.* Let $\{\theta_1, \cdots, \theta_K, \cdots, \theta_d\}$ be a set of orthonormal basis in $\mathcal{R}^d$ where the first $K$ vectors are $\theta_1, \cdots, \theta_K$, respectively. Let $R_l = [\theta_1, \cdots, \theta_d]$ be an orthogonal matrix whose i-th column vector is $\theta_i$, $U_l = [\theta_{K+1}, \cdots, \theta_d]$. Since $A_l = [\theta_1, \cdots, \theta_K]$, we have $R_l = [A_l, U_l]$ (the concatenation of columns of $A$ and $U$). Let $\mathbf{I}^{d-K} = [\text{id}_1, \cdots, \text{id}_{d-K}]^T$ be a marginal transformation that consists of $d - K$ 1D identity transformation,

$$\hat{\Psi}_l = \begin{bmatrix} \Psi_l \\ \mathbf{I}^{d-K} \end{bmatrix}, \text{ we have}$$

$$\begin{aligned} X_{l+1} &= A_l \Psi_l(A_l^T X_l) + X_l - A_l A_l^T X_l \\ &= A_l \Psi_l(A_l^T X_l) + R_l R_l^T X_l - A_l A_l^T X_l \\ &= A_l \Psi_l(A_l^T X_l) + [A_l, U_l] \begin{bmatrix} A_l^T \\ U_l^T \end{bmatrix} X_l - A_l A_l^T X_l \\ &= A_l \Psi_l(A_l^T X_l) + U_l U_l^T X_l \\ &= A_l \Psi_l(A_l^T X_l) + U_l \mathbf{I}^{d-K}(U_l^T X_l) \\ &= [A_l, U_l] \begin{bmatrix} \Psi_l \\ I^{d-K} \end{bmatrix} ([A_l, U_l]^T X_l) \\ &= R_l \hat{\Psi}_l(R_l^T X_l). \end{aligned}$$

$$(17)$$

Since $R_l$ is an orthogonal matrix with determinant $\pm 1$, and the Jacobian of the marginal transformation $\hat{\Psi}_l$ is diagonal, the Jacobian determinant of the above equation can be written as

$$\begin{aligned} \det(\frac{\partial X_{l+1}}{\partial X_l}) &= \prod_{k=1}^{K} \frac{d\Psi_{lk}(x)}{dx} \cdot \prod_{k=1}^{d-K} \frac{d(\text{id}_k(x))}{dx} \\ &= \prod_{k=1}^{K} \frac{d\Psi_{lk}(x)}{dx}. \end{aligned}$$

$$(18)$$

$\square$

### C.2. Objective

At iteration $l$, the objective of SINF can be written as:

$$\mathcal{F}_l = \min_{\{\Psi_{l1}, \cdots, \Psi_{lK}\}} \max_{\{\theta_{l1}, \cdots, \theta_{lK}\} \text{ orthonormal}}$$

$$\left( \frac{1}{K} \sum_{k=1}^{K} W_p^p(\Psi_{lk}((\mathcal{R}p_{1,l})(\cdot, \theta_{lk})), (\mathcal{R}p_2)(\cdot, \theta_{lk})) \right)^{\frac{1}{p}}. \quad (19)$$

The algorithm firstly optimize $\theta_{lk}$ to maximize the objective, with $\Psi_{lk}$ fixed to identical transformations (equivalent to Equation 4). Then the axes $\theta_{lk}$ are fixed and the objective is minimized with marginal matching $\Psi_l$. The samples are updated, and this process repeats until convergence.

### C.3. Monotonic Rational Quadratic Spline

Monotonic Rational Quadratic Splines (Gregory & Delbourgo, 1982; Durkan et al., 2019) approximate the function in each bin with the quotient of two quadratic polynomials. They are monotonic, contineously differentiable, and can be inverted analytically. The splines are parametrized by the coordinates and derivatives of $M$ knots: $\{(x_m, y_m, y_m')\}_{m=1}^{M}$, with $x_{m+1} > x_m$, $y_{m+1} > y_m$ and $y_m' > 0$. Given these parameters, the function in bin $m$ can be written as (Durkan et al., 2019)

$$y = y_m + (y_{m+1} - y_m) \frac{s_m \xi^2 + y_m' \xi(1-\xi)}{s_m + \sigma_m \xi(1-\xi)}, \qquad (20)$$

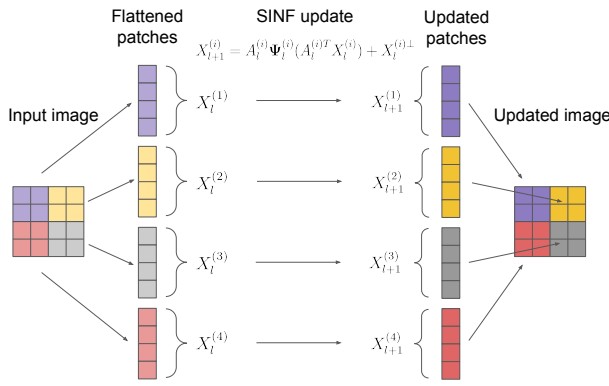

Flattened patches    SINF update    Updated patches
$X_{l+1}^{(i)} = A_l^{(i)} \Psi_l^{(i)} (A_l^{(i)T} X_l^{(i)}) + X_l^{(i)\perp}$

*Figure 4.* Illustration of the patch-based approach with $S = 4$, $p = 2$ and $q = 2$. At each iteration, different patches are modeled separately. The patches are randomly shifted in each iteration assuming periodic boundaries.

where $s_m = (y_{m+1} - y_m)/(x_{m+1} - x_m)$, $\sigma_m = y'_{m+1} + y'_m - 2s_m$ and $\xi = (x - x_m)/(x_{m+1} - x_m)$. The derivative is given by

$$\frac{dy}{dx} = \frac{s_m^2[y'_{m+1}\xi^2 + 2s_m\xi(1-\xi) + y'_m(1-\xi)^2]}{[s_m + \sigma_m\xi(1-\xi)]^2}. \quad (21)$$

Finally, the inverse can be calculated with

$$x = x_m + (x_{m+1} - x_m)\frac{2c}{-b - \sqrt{b^2 - 4ac}}, \quad (22)$$

where $a = (s_m - y'_m) + \zeta\sigma_m$, $b = y'_m - \zeta\sigma_m$, $c = -s_m\zeta$ and $\zeta = (y - y_m)/(y_{m+1} - y_m)$. The derivation of these formula can be found in Appendix A of Durkan et al. (2019).

In our algorithm the coordinates of the knots are determined by the quantiles of the marginalized PDF (see Algorithm 1). The derivative $y'_m$ ($1 < m < M$) is determined by fitting a local quadratic polynomial to the neighboring knots $(x_{m-1}, y_{m-1})$, $(x_m, y_m)$, and $(x_{m+1}, y_{m+1})$:

$$y'_m = \frac{s_{m-1}(x_{m+1} - x_m) + s_m(x_m - x_{m-1})}{x_{m+1} - x_{m-1}}. \quad (23)$$

The function outside $[x_1, x_M]$ is linearly extrapolated with slopes $y'_1$ and $y'_M$. In SIG, $y'_1$ and $y'_M$ are fixed to 1, while in GIS they are fitted to the samples that fall outside $[x_1, x_M]$.

We use $M = 400$ knots in SIG to interpolate each $\Psi_{l,k}$, while in GIS we set $M = \min(\sqrt{N_{\text{train}}}, 200)$. The performance is insensitive to these choices, as long as $M$ is large enough to fully characterize the 1D transformation $\Psi_{l,k}$.

### C.4. Patch-based hierarchical approach for SIG

Generally speaking, the neighboring pixels in images have stronger correlations than pixels that are far apart. This fact has been taken advantage by convolutional neural networks, which outperform Fully Connected Neural Networks

$q = 8, p = 1$ $\quad q = 4, p = 2$ $\quad q = 2, p = 4$

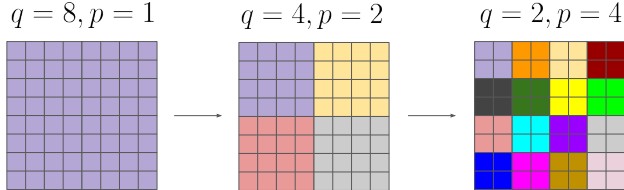

*Figure 5.* Illustration of the hierarchical modeling of an $S = 8$ image. The patch size starts from $q = 8$ and gradually decreases to $q = 2$.

(FCNNs) and have become standard building blocks in computer vision tasks. Like FCNNs, vanilla SIG and GIS make no assumption about the structure of the data and cannot model high dimensional images very well. Recently, Meng et al. (2020) proposed a patch-based approach, providing a different way to improve the modeling of the local correlations for NF models. The patch-based approach decomposes an $S \times S$ image into $p \times p$ patches, with $q \times q$ neighboring pixels in each patch ($S = pq$). In each iteration the marginalized distribution of each patch is modeled separately without considering the correlations between different patches. This approach effectively reduces the dimensionality from $S^2$ to $q^2$, at the cost of ignoring the long range correlations.

To reduce the effects of ignoring long range correlations, we propose a hierarchical model. In SIG, we start from modeling the entire images, which corresponds to $q = S$ and $p = 1$. After some iterations the samples show correct structures, indicating the long range correlations have been modeled well. We then gradually decrease the patch size $q$ until $q = 2$, which allows us to gradually focus on the smaller scales. Assuming a periodic boundary condition, we let the patches randomly shift in each iteration. If the patch size $q$ does not divide $S$, we set $p = \lfloor S/q \rfloor$ and the rest of the pixels are kept unchanged.

### C.5. Regularization of GIS

We add regularization to GIS for density estimation tasks to further improve the performance and reduce overfitting. The regularization is added in the following two aspects:
1) The weight matrix $W_l$ is regularized by limiting the maximum number of iteration $J_{\text{maxiter}}$ (see Algorithm 2). We set $J_{\text{maxiter}} = N/d$, where $N$ is the number of training data and $d$ is the dimensionality. Thus for very small datasets ($N/d \to 1$) the axes of marginal transformation are almost random. This has no effect on datasets of regular size.
2) The CDFs in Equation 6 are estimated using KDE, and the 1D marginal transformation is regularized with:

$$\tilde{\psi}_{l,k}(x) = (1 - \alpha)\psi_{l,k}(x) + \alpha x, \quad (24)$$

where $\alpha \in [0, 1)$ is the regularization parameter. $\tilde{\psi}_{l,k}$ is the regularized transformation. As $\alpha$ increases, the performance

improves, but more iterations are needed to converge. Thus $\alpha$ controls the trade-off between performance and speed.

## D. Related work

Iterative normalizing flow models called RBIG (Chen & Gopinath, 2000; Laparra et al., 2011) are simplified versions of GIS, as they are based on a succession of rotations followed by 1D marginal Gaussianizations. Iterative Distribution Transfer (IDT) (Pitié et al., 2007) is a similar algorithm but does not require the base distribution to be a Gaussian. These models do not scale well to high dimensions because they do not have a good way of choosing the axes, and they are not competitive against modern NFs trained on $p(x)$ (Meng et al., 2020). Meng et al. (2019) use a similar algorithm Projection Pursuit Monge Map (PPMM) to construct OT maps. They propose to find the most informative axis using Projection Pursuit (PP) in each iteration, and show that PPMM works well in low-dimensional bottleneck settings ($d = 8$). However, it has yet to be proven that PPMM scales to high dimensions, considering that PP scales as $\mathcal{O}(d^3)$. A DL, non-iterative version of these models is Gaussianization Flow (GF) (Meng et al., 2020), which trains on $p(x)$ and achieves good density estimation results in low dimensions, but does not have good sampling properties in high dimensions. RBIG, GIS and GF have similar architectures but are trained differently. We compare their density estimation results in Section 4.2.

Another iterative generative model is Sliced Wasserstein Flow (SWF) (Liutkus et al., 2019). Similar to SIG, SWF tries to minimize the SWD between the distributions of samples and the data, and transforms this problem into solving a d dimensional PDE. The PDE is solved iteratively by doing a gradient flow in the Wasserstein space, and they show SWF works well for low dimensional bottleneck features. However, in each iteration the algorithm requires evaluating an integral over the $d$ dimensional unit sphere approximated with Monte Carlo integration, which does not scale well to high dimensions. Another difference with SIG is that SWF does not have a flow structure, cannot be inverted, and does not provide the likelihood. We compare the sample qualities between SWF and SIG in Section 4.1.

SWD, max SWD and other slice-based distance (e.g. Cramér-Wold distance) have been widely used in training generative models (Deshpande et al., 2018; 2019; Wu et al., 2019; Kolouri et al., 2018; Knop et al., 2018; Nguyen et al., 2020b;a; Nadjahi et al., 2020). Wu et al. (2019) propose a differentiable SWD block composed of a rotation followed by marginalized Gaussianizations, but unlike RBIG, the rotation matrix is trained in an end-to-end DL fashion. They propose Sliced Wasserstein AutoEncoder (SWAE) by adding SWD blocks to an AE to regularize the latent variables, and show that its sample quality outperforms

VAE and AE + RBIG. Nguyen et al. (2020b;a) generalize the max-sliced approach using parametrized distributions over projection axes. Nguyen et al. (2020b) propose Mixture Spherical Sliced Fused Gromov Wasserstein (MSSFG), which samples the slice axes around a few informative directions following Von Mises-Fisher distribution. They apply MSSFG to training Deterministic Relational regularized AutoEncoder (DRAE) and name it mixture spherical DRAE (ms-DRAE). Nguyen et al. (2020a) go further and propose Distributional Sliced Wasserstein distance (DSW), which tries to find the optimal axes distribution by parametrizing it with a neural network. They apply DSW to the training of GANs, and we will refer to their model as DSWGAN in this paper. We compare the sample qualities between SWAE and SIG in Section 4.1. Besides GANs and AE-based models, OT has also been used in constructing NFs (Zhang et al., 2018; Finlay et al., 2020b; Onken et al., 2020; Finlay et al., 2020a; Huang et al., 2020).

Grover et al. (2018) propose Flow-GAN using a NF as the generator of a GAN, so the model can perform likelihood evaluation, and allows both maximum likelihood and adversarial training. Similar to our work they find that adversarial training gives good samples but poor $p(x)$, while training by maximum likelihood results in bad samples. Similar to SIG, the adversarial version of Flow-GAN minimizes the Wasserstein distance between samples and data, and has a NF structure. We compare their samples in Section 4.1.

## E. Hyperparameter study and ablation analysis

Here we study the sensitivity of SINF to hyperparameters and perform ablation analyses.

### E.1. Hyperparameter $K$, objective function, and patch based approach

We firstly test the convergence of SIG on MNIST dataset with different $K$ choices. We measure the SWD (Equation 2) and max SWD (Equation 3) between the test data and model samples for different iterations (without patch based hierarchical modeling). The results are presented in Figure 6. The SWD is measured with 10000 Monte Carlo samples and averaged over 10 times. The max SWD is measured with Algorithm 2 ($K = 1$) using different starting points in order to find the global maximum. We also measure the SWD and max SWD between the training data and test data, which gives an estimate of the noise level arising from the finite number of test data. For the range of $K$ we consider ($1 \leq K \leq 128$), all tests we perform converges to the noise level, and the convergence is insensitive to the choice of $K$, but mostly depends on the total number of 1D transformations ($N_{\text{iter}} \cdot K$). As a comparison, we also try running SIG with random orthogonal axes per iteration, and

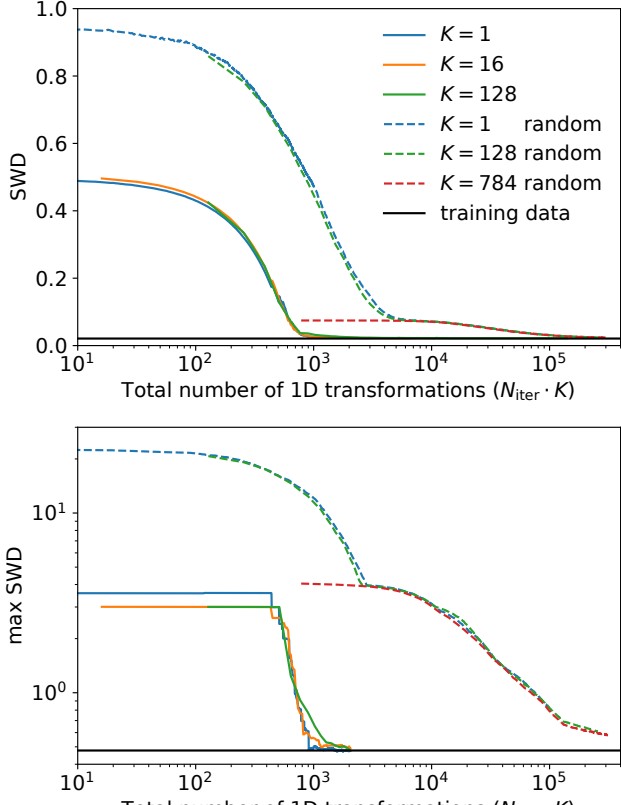

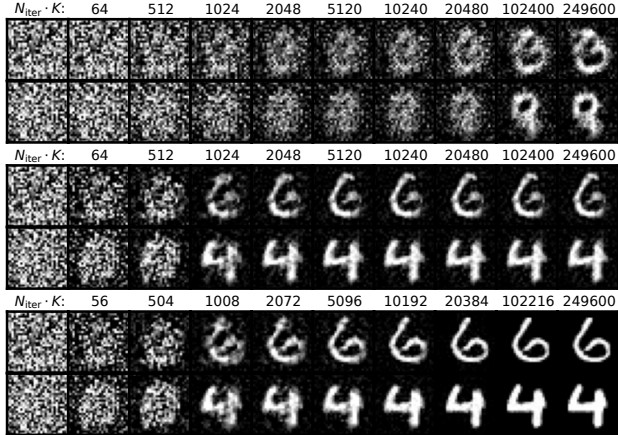

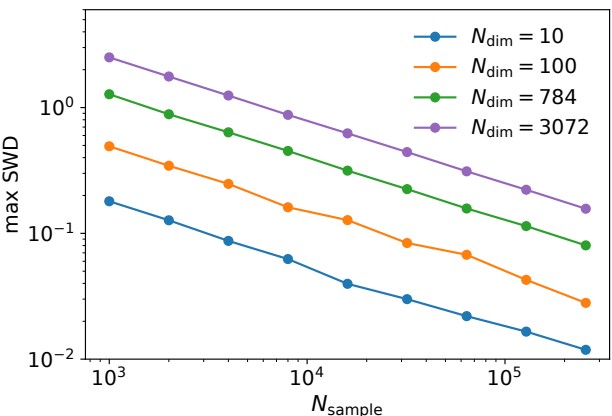

*Figure 7.* Top panel: SIG samples with random axes ($K = 64$). Middle panel: SIG samples with optimized axes ($K = 64$). Bottom panel: SIG samples with optimized axes and patch based hierarchical approach. The numbers above each panel indicate the number of marginal transformations.

*Figure 6.* Sliced Wasserstein Distance (SWD, top panel) and Max-Sliced Wasserstein Distance (max SWD, bottom panel) between the MNIST test data and model samples as a function of total number of marginal transformations. The legend in the top panel also applies to the bottom panel. The SWD and max SWD between the training data and test data is shown in the horizontal solid black lines. The lines with "random" indicate that the axes are randomly chosen (like RBIG) instead of using the axes of max K-SWD. We also test $K = 2$, $4$, $8$, $32$, and $64$. Their curves overlap with $K = 1$, $16$ and $128$ and are not shown in the plot.

*Figure 8.* The measured maximum sliced Wasserstein distance between two Gaussian datasets as a function of number of samples. 10 different starting points are used to find the global maximum.

for MNIST, our greedy algorithm converges with two orders of magnitude fewer marginal transformations than random orthogonal axes (Figure 6).

For $K = 1$, the objective function (Equation 19) is the same as max SWD, so one would expect that the max SWD between the data and the model distribution keep decreasing as the iteration number increases. For $K > 1$, the max K-SWD is bounded by max SWD (Equation 12 and 13) so one would also expect similar behavior. However, from Figure 6 we find that max SWD stays constant in the first 400 iterations. This is because SIG fails to find the global maximum of the objective function in those iterations, i.e., the algorithm converges at some local maximum that is almost perpendicular to the global maximum in the high dimensional space, and therefore the max SWD is almost unchanged. This suggests that our algorithm does not re-

quire global optimization of $A$ at each iteration: even if we find only a local maximum, it can be compensated with subsequent iterations. Therefore our model is insensitive to the initialization and random seeds. This is very different from the standard non-convex loss function optimization in deep learning with a fixed number of layers, where the random seeds often make a big difference (Lucic et al., 2018).

In Figure 7 we show the samples of SIG of random axes, optimized axes and hierarchical approach. On the one hand, the sample quality of SIG with optimized axes is better than that of random axes, suggesting that our proposed objective max K-SWD improves both the efficiency and the accuracy of the modeling. On the other hand, SIG with optimized axes has reached the noise level on both SWD and max SWD at around 2000 marginal transformations

(Figure 6), but the samples are not good at that point, and further increasing the number of 1D transformations from 2000 to 200000 does not significantly improve the sample quality. At this stage the objective function of Equation 19 is dominated by the noise from finite sample size, and the optimized axes are nearly random, which significantly limits the efficiency of our algorithm. To better understand this noise, we do a simple experiment by sampling two sets of samples from the standard normal distribution $\mathcal{N}(0, I)$ and measuring the max SWD using the samples. The true distance should be zero, and any nonzero value is caused by the finite number of samples. In Figure 8 we show the measured max SWD as a function of sample size and dimensionality. For small number of samples and high dimensionality, the measured max SWD is quite large, suggesting that we can easily find an axis where the marginalized PDF of the two sets of samples are significantly different, while their underlying distribution are actually the same. Because of this sample noise, once the generated and the target distribution are close to each other (the max K-SWD reached the noise level), the optimized axes becomes random and the algorithm becomes inefficient. To reduce the noise level, one needs to either increase the size of training data or decrease the dimensionality of the problem. The former can be achieved with data augmentation. In this study we adopt the second approach, i.e., we effectively reduce the dimensionality of the modeling with a patch based hierarchical approach. The corresponding samples are shown in the bottom panel of Figure 7. We see that the sample quality keeps improving after 2000 marginal transformations, because the patch based approach reduces the effective noise level.

### E.2. Effects of regularization parameter $\alpha$ in density estimation

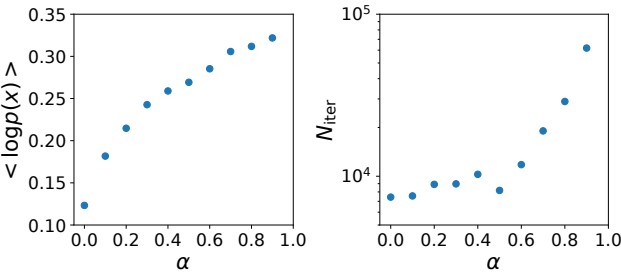

*Figure 9.* Test log-likelihood (left panel) and number of iterations (right panel) as a function of regularization parameter $\alpha$ on POWER dataset.

To explore the effect of regularization parameter $\alpha$, we train GIS on POWER dataset with different $\alpha$. We keep adding iterations until the log-likelihood of validation set stops improving. The final test $\log p$ and the number of iterations are shown in Figure 9. We see that with a larger $\alpha$, the algorithm gets better density estimation performance, at the cost of taking more iterations to converge. Setting the regu-

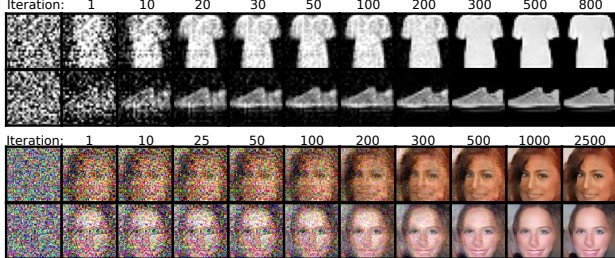

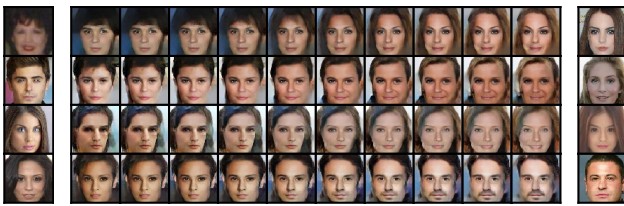

*Figure 10.* Gaussian noise (first column), Fashion-MNIST (top panel) and CelebA (bottom) samples at different iterations.

*Figure 11.* Middle: interpolations between CelebA samples from SIG. Left and right: the corresponding nearest training data.

larization parameter $\alpha$ is a trade-off between performance and computational cost.

## F. Other experiments

### F.1. Density estimation of full datasets

We perform density estimation with GIS on four UCI datasets (Lichman et al., 2013) and BSDS300 (Martin et al., 2001), as well as image datasets MNIST (LeCun et al., 1998) and Fashion-MNIST (Xiao et al., 2017). The data preprocessing of UCI datasets and BSDS300 follows Papamakarios et al. (2017). In Table 4 we compare our results with RBIG (Laparra et al., 2011) and GF (Meng et al., 2020). The former can be seen as GIS with random axes to apply 1D gaussianization, while the latter can be seen as training non-iterative GIS with MLE training on $p(x)$. We also list other NF models Real NVP (Dinh et al., 2017), Glow (Kingma & Dhariwal, 2018), FFJORD (Grathwohl et al., 2019), MAF (Papamakarios et al., 2017) and RQ-NSF (AR)(Durkan et al., 2019) for comparison.

We observe that RBIG performs significantly worse than current SOTA. GIS outperforms RBIG and is the first iterative algorithm that achieves comparable performance compared to maximum likelihood models. This is even more impressive given that GIS is not trained on $p(x)$, yet it outperforms GF on $p(x)$ on GAS, BSDS300 and Fashion-MNIST.

### F.2. More samples from SIG

In Figure 10 we show the SIG samples at different iterations. In Figure 11 we display interpolations between SIG samples, and the nearest training data, to verify we are not

*Table 4.* Negative test log-likelihood for tabular datasets measured in nats, and image datasets measured in bits/dim (lower is better).

| | Method | POWER | GAS | HEPMASS | MINIBOONE | BSDS300 | MNIST | Fashion |
|---|---|---|---|---|---|---|---|---|
| iterative | RBIG | 1.02 | 0.05 | 24.59 | 25.41 | -115.96 | 1.71 | 4.46 |
| | GIS (this work) | -0.32 | -10.30 | 19.00 | 14.26 | -155.75 | 1.34 | 3.22 |
| maximum likelihood | GF | -0.57 | -10.13 | 17.59 | 10.32 | -152.82 | 1.29 | 3.35 |
| | Real NVP | -0.17 | -8.33 | 18.71 | 13.55 | -153.28 | 1.06 | 2.85 |
| | Glow | -0.17 | -8.15 | 18.92 | 11.35 | -155.07 | 1.05 | 2.95 |
| | FFJORD | -0.46 | -8.59 | 14.92 | 10.43 | -157.40 | 0.99 | - |
| | MAF | -0.30 | -10.08 | 17.39 | 11.68 | -156.36 | 1.89 | - |
| | RQ-NSF (AR) | -0.66 | -13.09 | 14.01 | 9.22 | -157.31 | - | - |

memorizing the training data.

### F.3. Out of Distribution (OoD) detection

*Table 5.* OoD detection accuracy quantified by the AUROC of data $p(x)$ trained on Fashion-MNIST.

| Method | MNIST | OMNIGLOT |
|---|---|---|
| SIG (this work) | **0.980** | **0.993** |
| GIS (this work) | 0.824 | 0.891 |
| PixelCNN++ | 0.089 | - |
| IWAE | 0.423 | 0.568 |

OoD detection with generative models has recently attracted a lot of attention, since the $\log p$ estimates of NF and VAE have been shown to be poor OoD detectors: different generative models can assign higher probabilities to OoD data than to In Distribution (InD) training data (Nalisnick et al., 2019). One combination of datasets for which this has been observed is Fashion-MNIST and MNIST, where a model trained on the former assigns higher density to the latter.

SINF does not train on the likelihood $p(x)$, which is an advantage for OoD. Likelihood is sensitive to the smallest variance directions (Ren et al., 2019): for example, a zero variance pixel leads to an infinite $p(x)$, and noise must be added to regularize it. But zero variance directions contain little or no information on the global structure of the image. SINF objective is more sensitive to the meaningful global structures that can separate between OoD and InD. Because the patch based approach ignores the long range correlations and results in bad OoD, we use vanilla SINF without patch based approach. We train the models on F-MNIST, and then evaluate anomaly detection on test data of MNIST and OMNIGLOT (Lake et al., 2015). In Table 5 we compare our results to maximum likelihood $p(x)$ models PixelCNN++(Salimans et al., 2017; Ren et al., 2019), and IWAE (Choi et al., 2018). Other models that perform well include VIB and WAIC (Choi et al., 2018),

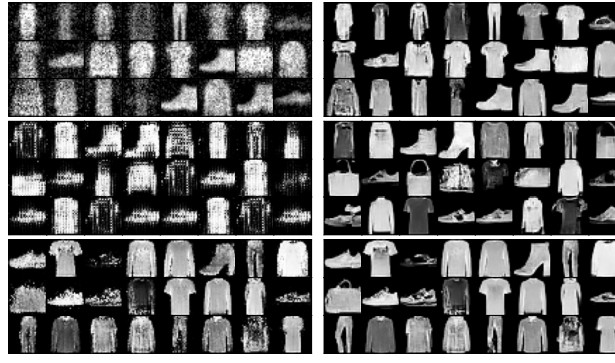

*Figure 12.* Fashion-MNIST samples before (left panel) and after SIG improvement (right panel). Top: SWF. Middle: Flow-GAN (ADV). Bottom: MAF.

which achieve 0.941, 0.943 and 0.766, 0.796, for MNIST and OMNIGLOT, respectively (below our SIG results). For the MNIST case Ren et al. (2019) obtained 0.996 using the likelihood ratio between the model and its perturbed version, but they require fine-tuning on some additional OoD dataset, which may not be available in OoD applications. Lower dimensional latent space PAE (Böhm & Seljak, 2020) achieves 0.997 and 0.981 for MNIST and OMNIGLOT, respectively, while VAE based likelihood regret (Xiao et al., 2020) achieves 0.988 on MNIST, but requires additional (expensive) processing.

### F.4. Improving the samples of other generative models

Since SIG is able to transform any distribution to the target distribution, it can also be used as a "Plug-and-Play" tool to improve the samples of other generative models. To demonstrate this, we train SWF, Flow-GAN(ADV) and MAF(5) on Fashion-MNIST with the default architectures in their papers, and then we apply 240 SIG iterations (30% of the total number of iterations in Section 4.1) to improve the samples. In Figure 12 we compare the samples before and after SIG improvement. Their FID scores improve from 207.6, 216.9 and 81.2 to 23.9, 21.2 and 16.6, respectively. These results can be further improved by adding more iterations.