# OpenReview forum: "Sliced Iterative Normalizing Flows"
_ICML.cc/2021/Workshop/INNF — INNF+ 2021 spotlighttalk_

### Official Review · Reviewer_A5v7 · 2021-06-10

**Rating:** Accept
**Confidence:** 4

**Summary:**

The paper propose a novel approach to construct Normalized Flows using Optimal Transport, by considering the Sliced Wasserstein (SW) distance. It relies on the insight that one could, in principle, approximate a probability distribution by aggregating 1-D Wasserstein distances between their projections  ("slices")  onto all directions of the sphere.

The computational method proposed does not use concepts such as mini-batching, stochastic gradient descent and gradient back-propagation through deep layers as used previously in Optimal Transport based Normalizing Flows.


**Justification For Rating:**

The paper has an original approach: (i) I am not aware of any use of Sliced Wasserstein (SW) distance in Normalizing Flows (NFs); (ii) the computational method proposed does not use concepts such as mini-batching, stochastic gradient descent and gradient back-propagation through deep layers as used previously in Optimal Transport (OT) based Normalizing Flows;

Moreover, the sample quality in Fig. 2 is nice compared to these obtained in the NFs literature. It is very unfortunate there is no comparison with results obtained in OT based methods for NFs as, for instance, in reference [6]. In fact, the authors acknowledge the relevance of OT in Generative Models as VAE and GANs, but completely ignores the use of OT in Flow-based Generative Models.

It would be nice to see a comparison of both (i) quality image generation and (ii) difference in the flow approximation among the Sliced Wasserstein approach proposed and the ones in [4,5,6].

In the conclusion, the authors makes a strong statement that “SIG is the first greedy deep learning algorithm that is competitive with the SOTA generators in high dimensions”. I am not confident on that and I would leave to the other reviewers to comment on it. Also, I have done a superficial reading on the supplementary material and have not checked all details.

My understanding is that the contributions are mainly at the computational level. It follows recent ideas on the use of Optimal Transport methods to construct Normalizing Flows, see references [1,3,5,6]. At the theoretical level, Sliced Wasserstein distance has been introduced already in 2011 (and studied from 2015). As acknowledge by the author(s), Sliced Wasserstein (or even the max-Sliced Wasserstein) have been widely used in training generative models in other contexts as, for instance, GANs.

Comments on style:

Suggestions:

For a better readability, I would suggest to introduce (for simplicity) a less general version of the Radon transform in the introduction by simply computing the projections as done in Cuturi&Peyre’’s book (Sliced Wasserstein Distance and Barycenters section). The authors could mention that the operator Radon transform is on appendix. It is not clear to me why the level of generally (infinite dimensional spaces) presented in the appendix is relevant in the main text.

Also, I would write in the main text where regularisation has been used to improve quality/overfitting as refer to the appendix for details.

Literature suggestions:

Recent literature on a OT-Flow-based Generative Models

[1]  Zhang, Linfeng, and Lei Wang. "Monge-amp\ere flow for generative modeling." arXiv preprint arXiv:1809.10188 (2018).

[2] Genevay, Aude, Gabriel Peyré, and Marco Cuturi. "Learning generative models with Sinkhorn divergences." International Conference on Artificial Intelligence and Statistics. PMLR, 2018.

[3] Finlay, Chris, et al. "How to train your neural ODE: the world of Jacobian and kinetic regularization." International Conference on Machine Learning. PMLR, 2020.

[4] Onken, Derek, et al. “OT-flow: Fast and accurate continuous normalizing flows via optimal transport." arXiv preprint arXiv:2006.00104 (2020).

[5] Finlay, Chris, et al. "Learning normalizing flows from Entropy-Kantorovich potentials." arXiv preprint arXiv:2006.06033 (2020).

[6] Huang, Chin-Wei, et al. "Convex Potential Flows: Universal Probability Distributions with Optimal Transport and Convex Optimization." arXiv preprint arXiv:2012.05942 (2020).


(Geniune) Questions:

1. The author(s) mention “SINF has very few hyperparameters, and is very insensitive to their choice”. Is this demonstrated along the text in which experiments? Where can we find this insight?
2. Algorithm 1 line 059: max K-SWD has three entries while the K-SWD defined in equation (4) has only two.  Also, I think in that line you are computing the argmax not the max.
3. At line 94 the authors mention an equivalence of “matching” the full distribution by considering the slices in all possible directions. How does it is realistic in practise? Did the authors think how the approximations done in SIF may affect this theoretical insight?
4. Following question 3: are the authors aware of an equivalence (if true) between the ‘exact’ Sliced Wasserstein flow and a Wasserstein geodesic at least in d=1,d=2? Notice that there are close-form solutions for the Wasserstein geodesic when both p_1 and p_2 are Gaussians, so one could compute the exact Wasserstein geodesic even in higher dimensions, see reference [C].

Other literature comments:
Up to my knowledge, the Sliced Wasserstein were introduced in [A,B] based on an idea of Marc Bernot.

[A] Rabin, Julien, et al. "Wasserstein Barycenter and its application to texture mixing." International Conference on Scale Space and Variational Methods in Computer Vision. Springer, Berlin, Heidelberg, 2011.

[B] Nicolas Bonneel, Julien Rabin, Gabriel Peyré, and Hanspeter Pfister. Sliced and Radon Wasserstein barycenters of measures. Journal of Mathematical Imaging and Vision, 51(1):22–45, 2015.

[C] Takatsu, Asuka. "On Wasserstein geometry of Gaussian measures." Probabilistic approach to geometry. Mathematical Society of Japan, 2010. 463-472.

---

### Official Review · Reviewer_x2q3 · 2021-06-12

**Rating:** Accept
**Confidence:** 4

**Summary:**

This work develops an iterative algorithm to transform an arbitrary probability distribution function (PDF) into the
target PDF. The proposed algorithm transforms the problem of matching a high dimensional distribution into a task of matching a series of 1D marginals, and then matches each marginal using iterative Optimal Transport. The authors apply this algorithm to building normalizing flows and show that the proposed method is able to generate high quality image samples, and obtain better results on small dataset density estimation tasks compared to NFs trained via maximum likelihood estimation.

**Justification For Rating:**

1. The paper is well written and easy to follow.
2. The proposed approach has theoretical guarantee, proofs and theoretical analysis are also provided (in appendix).
3. The experiment sections are very detailed and method comparisons are thorough.
4. The proposed method has strong empirical performances. The proposed method can generate high quality image samples, while many previous iterative algorithms were unable to produce good samples on high dimensional image datasets. The proposed method also achieves competitive likelihoods, achieving better performance than previous iterative algorithms on tabular datasets and image datasets.

---

### Official Review · Reviewer_Pgzu · 2021-06-13

**Rating:** Accept
**Confidence:** 3

**Summary:**

The paper introduces SINF: an algorithm for learning normalising flows from an arbitrary distribution to a distribution of interest by minimizing Wasserstein distances over 1-dimensional slices. The validity of the algorithm rests on the Carmer-Wold theorem that states that two functions are identical if all 1-dimensional slices are the same.

Interestinly, SINF iteratively optimises a sequence of projection axes and flows in batch mode, without using minibatches or gradient descent. At each iteration the algorithm searches for the K orthonormal axes that maximise the sliced Wasserstein distance and calculates the optimal transport program that matches the CDF of the transformed distribution and distribution of interest.

The authors detail two instantiations of SINF:  1) SIG that transforms samples from a multivariate-Gaussian distribution into the distribution of interest, and 2) GIS which iteratively transforms the dataset into samples from a multivariate-Gaussian.

SIG produces good samples as shown on 4 datasets (and measured by FID). GIS is reported to be more adequate for density estimation, which is shown in some UCI datasets and BSDS300, and seems most advantageous on the low data regime.


**Justification For Rating:**

The paper is well written, presents interesting and novel ideas, and has an adequate amount of theoretical and empirical results for a workshop.

A longer format version of the paper could benefit from:
* More insight on why GIS and SIG are adequate for density-estimation vs sample generation (which is mentioned in passing now on lines 151-156).
* Specification which non-GIS models collapsed and which didn't fit GPU memory on section 4.1.
* Showing samples at T=1 for all datasets. Or comments on the possible disadvantages of tuning temperatures (e.g. biased class sampling).
* Comments on whether it is possible to use SINF efficiently for conditional sampling.

---

### Decision · Program_Chairs · 2021-06-14

Accept (spotlight talk)